# Bactericidal and Bioresorbable Calcium Phosphate Cements Fabricated by Silver-Containing Tricalcium Phosphate Microspheres

**DOI:** 10.3390/ijms21113745

**Published:** 2020-05-26

**Authors:** Michiyo Honda, Yusuke Kawanobe, Kohei Nagata, Ken Ishii, Morio Matsumoto, Mamoru Aizawa

**Affiliations:** 1Department of Applied Chemistry, School of Science and Technology, Meiji University, Kanagawa 214-8571, Japan; ysk.k.0228@gmail.com (Y.K.); kohei.99991@gmail.com (K.N.); mamorua@meiji.ac.jp (M.A.); 2Department of Orthopedic Surgery, School of Medicine, International University of Health and Welfare Mita Hospital, 1-4-3, Mita, Minato-ku, Tokyo 108-8329, Japan; kenishii88@gmail.com; 3Department of Orthopaedic Surgery, School of Medicine, Keio University, Tokyo 160-8582, Japan; morio@a5.keio.jp

**Keywords:** calcium phosphate cement, silver-containing tricalcium phosphate, antimicrobial property, implant-related infection

## Abstract

Bacterial adhesion to the calcium phosphate surface is a serious problem in surgery. To prevent bacterial infection, the development of calcium-phosphate cements (CPCs) with bactericidal properties is indispensable. The aim of this study was to fabricate antibacterial CPCs and evaluate their biological properties. Silver-containing tricalcium phosphate (Ag-TCP) microspheres consisting of α/β-TCP phases were synthesized by an ultrasonic spray-pyrolysis technique. The powders prepared were mixed with the setting liquid to fabricate the CPCs. The resulting cements consisting of β-TCP and hydroxyapatite had a porous structure and wash-out resistance. Additionally, silver and calcium ions could be released into the culture medium from Ag-TCP cements for a long time accompanied by the dissolution of TCP. These data showed the bioresorbability of the Ag-TCP cement. In vitro antibacterial evaluation demonstrated that both released and immobilized silver suppressed the growth of bacteria and prevented bacterial adhesion to the surface of CPCs. Furthermore, histological evaluation by implantation of Ag-TCP cements into rabbit tibiae exhibited abundant bone apposition on the cement without inflammatory responses. These results showed that Ag-TCP cement has a good antibacterial property and good biocompatibility. The present Ag-TCP cements are promising for bone tissue engineering and may be used as antibacterial biomaterials.

## 1. Introduction

The similarity of calcium phosphate (CaP) to the mineral phase in bone contributes to good biocompatibility as well as excellent osteoconductivity. A variety of CaP biomaterials, such as blocks, granules, and cements, have been developed for tissue engineering. In particular, the advantage of calcium phosphate cement (CPC) is that its shape is easily controllable, which enables minimally invasive surgery, and it can be quickly integrated with our own bones to transform the matrix of the bone [1]. Additionally, CPC paste enables to self-set without inflammatory responses to the surrounding tissue by a combined dissolution–precipitation model [2]. Recently, the most commonly used injection has been poly-(methylmethacrylate) (PMMA), which is not degradable; however, its high curing temperatures can induce necrosis around the implanted tissues [3]. On the other hand, CPCs provide bioresorbability, biocompatibility, and osteoconductivity [4,5,6,7]. The CPCs were fabricated by mixing CaP powders and a suitable aqueous liquid form of hydroxyapatite (HAp) with low crystallinity upon setting [8]. The resulting cements, based on the reaction of dissolution and precipitation, were replaced by newly formed bone at body temperature and in a physiological environment. However, due to their low macroporosity, these cements take a long time to be reabsorbed via cell-mediated processes such as osteoclasts. To enhance the rapid resorption and concomitant osseointegration, various methods have been employed to build macropores [7,9,10]. Furthermore, bioresorbable CPCs that can set at low temperatures allow the incorporation of drugs and make them candidates as drug carriers [11]. CaPs, especially HAp and TCP, have attracted great attention in using bone substitute and drug delivery career simultaneously as potential applications for bone regeneration [12]. Indeed, bioresorbable CPCs loaded with antibiotics have been developed because implants including CPCs become potential sources of bacterial infection [13,14,15]. A local drug delivery system is an effective means to avoid the systemic toxicity and to deliver higher drug concentrations to infected bone. The most likely treatment of biofilm-based infections is to prevent the onset of infection by inhibiting the initial attachment stage [16]. Various approaches have been conducted to create antibacterial coatings by natural and synthetic materials. The antimicrobial activity of a surface can be accomplished through two different approaches: i) the release of a chemical or antibacterial agent from the surface of medical device that targets the surrounding bacteria; ii) immobilization of antibiotic molecules on the device surface that can inhibit the bacterial adhesion to the surface or kill the attached bacteria [17].

Major attention has been paid to antibiotics to inhibit infections that occur during surgical operations or treatment of common infections [11]. However, antimicrobial resistance remains a significant problem for our health. Antibiotic-resistant infections are ever-increasing, and pathogens acquire resistance to multiple drugs [18]. Several agents have been proposed to decrease the incidence of implant-associated infections [19,20,21]. Silver, a bactericidal agent, has been developed as an effective strategy to reduce bacterial cell adhesion and to inhibit biofilm formation. Silver exhibits broad-spectrum antibacterial activity against numerous strains of bacteria, fungi, and possibly some viruses, including some drug-resistant microbial strains [22]. It is well known that silver ions can deactivate cellular enzymes and DNA by coordinating with electron-donating groups such as thiols, carboxylates, amides, imidazoles, indoles, and hydroxyls [23]. Additionally, the Ag^+^ ions can pit on the bacterial cell surface and increase membrane permeability, resulting in cell death [24]. The interface with sulfhydryl groups in enzymes and proteins is associated with antibacterial action of silver ion. In addition, interaction with Ag^+^ and nucleic acid can prevent cell division and reproduction [25]. Numerous studies have been employed to examine silver-containing or silver-doped CaP implants [21,26,27,28,29]. In our previous investigation, Ag-HAp (silver-containing hydroxyapatite) powders were prepared with desired contents of silver ions via an ultrasonic spray-pyrolysis (USSP) route [30]. The Ag-HAp powders containing more than 5% Ag had a bactericidal activity in vitro and in vivo as well as biocompatibility. However, these powders had no capacity for self-setting and anti-washout, which are essential elements for cement paste. Therefore, in this study, we synthesized Ag-TCP powders and fabricated Ag-TCP cements based on the reaction of dissolution and precipitation with a self-setting property, anti-washout capability, and antimicrobial activity.

The purpose of this study was to fabricate Ag-TCP cements for examination of their bactericidal properties and biocompatibility and to demonstrate their effectiveness as antibacterial bone substitutes.

## 2. Results and Discussion

### 2.1. Characterization of Ag-TCP Powders

We previously reported that Ag-HAp powders containing more than 5% of Ag had a bactericidal activity and exhibited good biocompatibility in vivo [30]. Therefore, in this study, we prepared two kinds of Ag-TCP powders (Ag mol% of 0 and 5; Ag-TCP(0) and Ag-TCP(5)) using USSP. Figure 1 illustrates the X-ray diffractometry (XRD) patterns of Ag-TCP(0) and Ag-TCP(5) powders. Each Ag-TCP powder was composed of α-TCP and β-TCP. The ratio of α-TCP and β-TCP was found to be about 2:1 by the reference intensity ratio method. The crystallinity of Ag-TCP(5) was lower than that of Ag-TCP(0). An analysis by inductively coupled plasma atomic emission spectroscopy (ICP-AES) showed that the amount of Ag contained in Ag-TCP(5) was 4.64 mol%, which was approximately 90% of the amount used during preparation.

The scanning electron microscopy (SEM) micrographs of Ag-TCP powders showed that both Ag-TCP(0) and Ag-TCP(5) powders were composed of microspheres with diameters of 0.5–3.0 μm (Figure 2a,b). Regardless of the Ag content, each particle exhibited very similar morphologies.

On the other hand, the transmission electron microscopy (TEM) micrographs revealed that the cores of Ag-TCP powders were translucent (Figure 2c,d). Our data suggest that the powders were partly composed of hollow and spherical particles. These results were in agreement with the Ag-HAp powders.

### 2.2. Fabrication of Ag-TCP Cements and Their Properties

To fabricate Ag-TCP cements, the above Ag-TCP powders were mixed with the setting liquid (2.5 mass% Na_2_HPO_4_) at a powder/liquid ratio of 1/0.3 (g·cm^−3^). Figure 3a,b show the XRD patterns of the Ag-TCP(5) powder and cement, respectively. Comparison of Ag-TCP(5) powder and cement demonstrated that the powder consisted mainly of α-TCP αand β-TCP phases, while the cement mainly consisted of β-TCP and HAp phases. These results indicate that α-TCP in powder transformed into HAp under setting. In fact, we could find the morphological changes at the surface of each powder by the fabrication of cements. As shown in Figure 4, Ag-TCP(5) cement sets based on the dissolution and precipitation reaction formed on the plate-like HAp at the surface of the microspheres. The same morphological changes were also observed in Ag-TCP(0) cement. On the other hand, no apparent changes were seen in the ratio of β-TCP phase in Ag-TCP cement. These data suggest that β-TCP did not transfer to HAp by hydrolysis reaction under present conditions (at room temperature and powder/liquid ratio).

Spherical powders would provide porous structure in the cement, which encourages vascularization and bone ingrowth. However, the pore size was too small for cells to penetrate into the center part of cement. By using porogens, cement would be dissolved more quickly, and the tissue ingrowth would be enhanced. Moreover, cell penetration into cement would help the dissolution of bone and bone cement by osteoclasts and subsequently the drug release. Therefore, adding a porogenic agent, such as gelatin, mannitol, and chitosan, is effective to enhance interconnectivity of the porosity [31,32]. In fact, the highly porous CPCs can offer the opportunity for drug delivery carrier and contain different types of drugs, including antibiotics, antimicrobial peptides, and growth factors [33]. However, their poor mechanical performance of CPCs still limits their applicability.

On the other hand, wash-out resistance is also required to fabricate CPCs for clinical use. Low strength and the lack of ability to self-harden in situ limited its use. In our previous work, we also developed silver-containing HAp powders and demonstrated that contents above 5 mol% Ag-HAp disc possessed both antibacterial activity and biocompatibility [30]. However, Ag-HAp powders had no capacity for self-setting and anti-washout. To overcome these limitations, we proposed a cement with self-hardening, washout resistance, and bactericidal properties by using Ag-TCP powders. To evaluate the anti-washout property, fabricated cements were immersed in water after incubation at 37 °C for 24 h (Figure 5). As a result, the Ag-HAp disc [30] was dispersed in the water (Figure 5b), but the Ag-TCP (5) and the Ag-TCP(0) cements did not collapse in water (Figure 5a). As opposed to the Ag-HAp discs, the Ag-TCP cements obtained self-hardening and washout resistance by setting based on the dissolution–precipitation reaction. In addition, mechanical strength of cement is also an important factor for a clinical application. To explore the mechanical property of Ag-TCP cements, compressive strength was measured. The compressive strength achieved after setting was on average 13.6 (±2.1) MPa for Ag-TCP(0) and 10.3 (±1.5) MPa for Ag-TCP(5), which is in the range of the compressive strength of cancellous bone. The compressive strength of Ag-TCP(5) was slightly lower than that of Ag-TCP(0), though precise factors were not clarified. These data show that the improvement of mechanical properties of these cements should be required by addition of various chemicals and changes of microstructure of powders [34]. Furthermore, the bending and the tensile properties of CPCs also should be investigated for more accurate analysis of mechanical properties [35].

Next, to assess the in vitro release kinetics of Ag^+^ ions from cements, Ag-TCP(0) and Ag-TCP(5) cements were immersed in Luria-Bertani (LB) media for 3, 5, 7, 14, and 21 days. The cumulative Ag^+^ ions, Ca^2+^ ions, and PO_4_^3−^ ions were measured by ICP-AES (Figure 6). ICP-AES analysis showed that almost equal amounts of Ag^+^ ions (4.86 × 10^−3^ μg/mg for day 3, 5.37 × 10^−3^ μg/mg for day 5, 5.50 × 10^−3^ μg/mg for day 7, and 4.55 × 10^−3^ μg/mg for day 14) were released in the first 14 days, after which the release levels decreased slightly (Figure 6a). Furthermore, the cumulatively released Ca^2+^ and PO_4_^3−^ ions in the LB medium at each time point were investigated (Figure 6b,c). In the first three days, large amounts of Ca^2+^ and PO_4_^3−^ ions released from each sample. The concentration of Ca^2+^ and PO_4_^3−^ released from Ag-TCP(5) cement was higher than that of Ag-TCP(0), and this contributed to the low crystallinity of Ag-TCP(5). The end products mainly consisted of β-TCP and HAp. Although sintered HAp is more stable, CPCs with low crystallinity fabricated in this study could be dissolved and released Ca^2+^, PO_4_^3−^, and Ag^+^ ions for a long term. In vivo situation, they can be degraded by different two ways as follows: (i) active resorption controlled by macrophages and osteoclasts; (ii) passive resorption thorough chemical dissolution or hydrolysis in body fluid.

### 2.3. Antibacterial Properties

To provide the localized antibiotic protection to bone defects, the drug-modified pre-mixed carriers for clinical practices concern the prevention of post-operative bone infections during implantation. Antibiotic resistance is a universal problem for treatment of patients with contaminated medical devices. To inhibit and treat biofilm-associated infections, a great deal of research has been conducted searching for alternative approaches. In this study, to examine the antibacterial activity of Ag-TCP cements, inhibition zone assay was performed using *Staphylococcus aureus* (*S. aureus*) (Figure 7). Inhibition zone was observed only around Ag-TCP(5) cements (Figure 7a). The area was quantitatively analyzed using Image J software (Figure 7b). The values of relative inhibition zone were close to Ag-HAp discs with Ag contents of 5 mol% [30]. These data indicate that released Ag^+^ ions from Ag-TCP(5) cement contributed the antibacterial activity. Furthermore, to investigate the reaction of *S. aureus* strains to Ag-TCP(5) cements and Ag-TCP(0) cements used as control, bacterial adhesion was visualized by LIVE/DEAD staining (Figure 8). As a result, the presence of viable cells (stained green) could be seen on the Ag-TCP(0) cements, and there were very few dead cells. On the other hand, very few viable cells and dead cells could be seen on the Ag-TCP(5) cements. These data suggest that the bacterial adherence to the surface of Ag-TCP(5) cement was inhibited by released Ag^+^ ions from Ag-TCP(5) cements. Interestingly, the aggregation of dead *S. aureus* was seen at various locations (Figure 8, Ag-TCP(5)).

To analyze the accumulation of *S. aureus* at the surface of Ag-TCP(5) cements, microstructure of cement and morphology of bacteria were observed by SEM (Figure 9). The images of SEM display the needle-like structures at the surface of each cement, and *S. aureus* could adhere on Ag-TCP(0) cement with normal morphology (Figure 9a). However, cells were covered with depositions on the Ag-TCP(5) cements (Figure 9b). To identify the depositions on Ag-TCP(5) cements, energy dispersive X-ray spectrometry (EDX) analysis was performed (Figure 9c). The EDX spectrum of Ag-TCP(5) cement demonstrated by asterisk in Figure 9b identified those depositions as silver in Figure 9c by the sharp Ag peaks. These data imply that the growth of bacteria was suppressed by immobilized silver as well as released silver ions. In the present investigation, bacteria were mainly killed by released Ag^+^ ions from Ag-TCP cement. On the other hand, we could observe that silver particles of Ag-TCP cement might have anchored to the bacterial membrane (Figure 9b). Physical changes in the bacterial cell wall such as membrane damage would occur, resulting in the leakage of cellular contents and bacterial death. As described above, Ag-TCP cement would have two antibacterial mechanisms.

### 2.4. In Vivo Biocompatibility

High concentrations of metal ions, including silver, can cause local toxicity. To evaluate the biocompatibility of Ag-TCP cements, histological examinations were performed using a rabbit model (Figure 10). Tibiae staining using Villanueva bone staining showed no abnormal pathologic findings in either Ag-TCP(0) or Ag-TCP(5) cement. We observed bone tissue-to-Ag-TCP cement direct bonding without causing damage to surrounding tissues. Villanueva bone staining showed that bone formation was observed around the Ag-TCP cements in the longitudinal sections of the implant in both groups. However, there were no differences in the rate of new bone formation in each cement. Taken together, Ag-TCP(5) cement has a good biocompatibility.

Ag^+^ ions disrupt cellular functions and attack multiple targets within cells to inactivate critical physiological roles. Release kinetics show that Ag^+^ ions constantly released from Ag-TCP cement were about 1.2 ppm, which was sustained for 3 weeks (Figure 6A). In our previous studies, no significant reduction of osteoblastic cell number was detected under 1.3 ppm of Ag^+^ ions [30,36]. Furthermore, continuous release of Ag^+^ ions was observed in Ag-TCP cement compared with Ag-HAp. This contributed to the dissolution of Ag-TCP cement. Taken together, biodegradable Ag-TCP cement shows the good biocompatibility as well as good antibacterial properties. However, weak mechanical properties have limited wider applications in hard tissue implants. The mechanical properties of the Ag-TCP cement should be improved for applying to load-bearing device.

Although CPC has no antimicrobial activity, it has been effective for bone repair. Therefore, they are usually used by loading antibiotics. The relatively fast drug release from CPCs should be controlled. In the case of this study, dissolution of Ag-TCP cements resulted in sustained release of Ag^+^. Furthermore, an effective concentration to inhibit biofilm formation for a longer duration could be allowed by the continuous slow release of Ag^+^. The release profiles of Ag^+^ ion from the cement would show the prevention of infection at the initial stage after implantation. The release kinetics of Ag^+^ ion was close to those of Ca^2+^ and PO_4_^3−^ ions. The bone remodeling and bone healing would enhance the effective antimicrobial activity. However, the precise mechanism of ion substitution was not clarified. We should examine the incorporation of Ag^+^ into the crystal lattice of the CaP. In addition, we must clarify “substituted-CaPs” or ‘‘doped-CaPs”, where the foreign ion is just adsorbed on the surface of the crystals.

## 3. Materials and Methods

### 3.1. Preparation of Ag-TCP Powders and Fabrication of Ag-TCP Cements

As described in our previous reports [30,37], the apparatus of the USSP system was composed of an atomizer, a heating zone (mullite tube: an ID of 2.5 cm and a height of 1 m; electric furnace: an ID of 3 cm and a height of 60 cm), a powder collecting zone (test-tube filter), and a controller. There were two electric furnaces in heating zone: (i) evaporation of the solvent from the droplets (a lower furnace, temperature: 300 °C); (ii) pyrolysis (an upper furnace, temperature: 850 °C). An aspirator (suction rate: 1.5 dm^3^·min^−1^) was used for introduction of droplets into the heating zone.

As listed in Table 1, Ca(NO_3_)_2_·4H_2_O, (NH_4_)_2_HPO_4_, HNO_3_, and AgNO_3_ were mixed for preparation of the starting solutions. Ag-TCP powders were obtained by spray-pyrolysis of the starting solution using the ultrasonic vibrator (Frequency: 2.4 MHz). The excess nitrate in as-prepared powders was removed by washing with ultrapure water (hereafter, washed powders). For fabrication of cements, 0.30 g of Ag-TCP powders were mixed with the setting liquid (2.5 mass% Na_2_HPO_4_) at a P/L (powder/liquid) ratio of 1/0.3 (g·cm^−3^). The resulting pastes were uniaxially compressed at 20 MPa to form cement specimens for antibacterial tests (15 mm diameter and 1–2 mm height) and biocompatibility tests (4 mm diameter and 8 mm height).

### 3.2. Characterization of Ag-TCP Powders and Ag-TCP Cements

The X-ray diffraction (XRD) was used to identify the crystalline phases of the resulting powders and conducted by the powder X-ray diffractometry (MiniFlex, Rigaku Co., Tokyo, Japan) with Cu-Kα radiation operating at 30 kV and 15 mA. Data were collected in the 2θ range of 4–50° with a step size of 0.04° and at a speed of 4°/min. The crystalline phase was identified by reference to the ICDD (International Center for Diffraction Data) database for α-, β-TCP, and HAp (#09-0348, #09-0169, and #9-0432). The scanning electron microscope (SEM; JSM-6390LA, JEOL, Tokyo, Japan) was used to observe the morphology of the sample powders and cements at an accelerating voltage of 15 kV. For SEM/energy dispersive X-ray spectrometry (EDX) measurement, samples were prepared by coating with Pt using sputtering prior to observation by SEM. Analysis of calcium, phosphorous, and silver in the resulting cement specimens was carried out using EDX. Observation of ultrastructure of resulting powders was performed by high-resolution transmission electron microscopy (HR-TEM; JEM-2100F, JEOL Tokyo, Japan) at an accelerated voltage of 200 kV. For preparation of HR-TEM samples, the powders were dispersed in ethanol and collected onto 300-mesh cupper grids coated with carbon (Agar Scientific, England). The contents of Ag^+^, Ca^2+^, and PO_4_^3−^ ions in the samples were measured using inductively coupled plasma atomic emission spectroscopy (ICP-AES; SPS7800, SII NanoTechnology, Tokyo, Japan). Compressive strength was measured by forming the cement specimens (a cylindrical shape of 5 mm in diameter and 6 mm in height) using a universal machine (Autograph AGS-X, Shimadzu, Kyoto, Japan) with a 5 kN load cell at a crosshead speed of 500 μm·min^−1^.

### 3.3. Release Kinetics of Ag^+^, Ca^2+^, and PO_4_^3−^ Ions from Ag-TCP Cements

Ag-TCP(0) and Ag-TCP(5) cements were set on 24 well polystyrene plates, which were immersed in 1.2 cm^3^ Luria-Bertani (LB) medium (Wako, Osaka, Japan) for desired period at 37 °C in an incubator. Medium was changed before the measurement of every ion (at day 3, 5, 7, 14, and 21). The concentrations of Ag^+^, Ca^2+^, and PO_4_^3−^ ions in medium from each Ag-TCP cement were then analyzed using the ICP-AES.

### 3.4. Antimicrobial Susceptibility Test

Gram-positive bacterium *Staphylococcus aureus* (*S. aureus*, IAM 1011) was used for antimicrobial susceptibility tests. *S. aureus* were cultured in nutrient broth and agar (Wako, Osaka, Japan) at 37 °C in an incubator.

#### 3.4.1. Zone of Inhibition Test

Zone of inhibition test was conducted for qualitative analysis of the bactericidal properties of the antimicrobial agents to inhibit bacterial growth onto the agar media [38].

Bacterial cultures (10^6^ CFU·cm^−3^) were spread on thin agar (10 cm^3^) in petri dishes. The cements were applied to the center of the thin agar plate. The additional agar (4 cm^3^) was overlaid onto the solidified nutrient agar. The agar plates were inoculated for 48 h at 37 °C. The size of “inhibition zone (*A*)” as given by Equation (1) where *D*_1_ and *D*_2_ were the area of inhibition zone and cement specimens, respectively.
*A* = (*D*_1_ − *D*_2_)/*D*_2_(1)

#### 3.4.2. Bacterial Viability Assay

For analysis of cell viability, samples were inoculated with bacterial suspensions for 24 h and stained with a BacLight LIVE/DEAD stain kit (Thermo, Waltham, MA, USA). In brief, supernatants were removed from the well, and the specimens were washed twice with phosphate-buffered saline (PBS). According to manufacturer’s instruction, the solution of LIVE/DEAD stain was prepared by mixing equal amounts of SYTO^®^ 9 and propidium iodide (PI) in PBS. Samples were then incubated for 15 min in the dark at room temperature. The stained bacteria on the cements were observed by fluorescence microscope (IX71, Olympus, Tokyo, Japan).

### 3.5. In Vivo Biocompatibility Test

In vivo tests were performed using male Japan White rabbits (about 16 weeks age, ~3.0 kg) for assessment of the biocompatibility and the osteoconductivity of Ag-TCP cements. The Ag-TCP cement specimens were fabricated by mixing Ag-TCP(0) or (5) powder and 2.5 mass% Na_2_HPO_4_ at a powder/liquid ratio of 1/0.3 (g∙cm^−3^) and packing into a cylindrical stainless mold. The resulting cement specimens (4 mm in diameter and 8 mm in height) were kept in air at room temperature for 24 h. The cement specimens were sterilized by Ethylene oxide gas.

The operation was performed under general anesthesia. A 4.4 mm (diameter) × 8 mm (depth) defects were generated in an epiphysis of the tibia. Ag-TCP cements were implanted into the defects for 8 weeks. Rabbits were sacrificed using sodium pentobarbital at week 8 following implantation, and specimens together with surrounding tissue were removed. Undecalcified histological samples were fixed in 70% ethanol and then dehydrated in an alcohol solution, defatted, embedded in methyl-methacrylate resin, and cut into sections using a microtome. The specimens were stained with Villanueva bone stain. Histological observation of sections was carried out by microscope (IX71, Olympus, Tokyo, Japan). All experiments were approved by the Keio University Institutional Animal Care and Use Committee (09067-(10),2016/11/14).

## 4. Conclusions

In this study, we prepared spherical Ag-TCP powder composed of α-TCP and β-TCP. Ag-TCP cements were fabricated using Ag-TCP powders and a setting liquid (2.5 mass% Na_2_HPO_4_) based on the dissolution and precipitation reaction. The cements provided a porous structure and anti-washout properties. The in vitro release kinetics of Ag^+^ ions from cements demonstrated that Ag-TCP(5) cement could release Ag^+^ ions for an extended duration accompanied by the dissolution of TCP. The antibacterial activity and biocompatibility of Ag-TCP cements were demonstrated in vitro and in vivo. In vitro, Ag-TCP(5) cement significantly inhibited the adhesion and the proliferation of *S. aureus*. On the other hand, the biocompatibility in vivo showed that there were no abnormal pathologic changes, and we could observe the direct bonding of the newly formed bone with cement specimens. Therefore, Ag-TCP cements are promising for bone tissue engineering and may be used as antibacterial biomaterials.

## Figures and Tables

**Figure 1 ijms-21-03745-f001:**
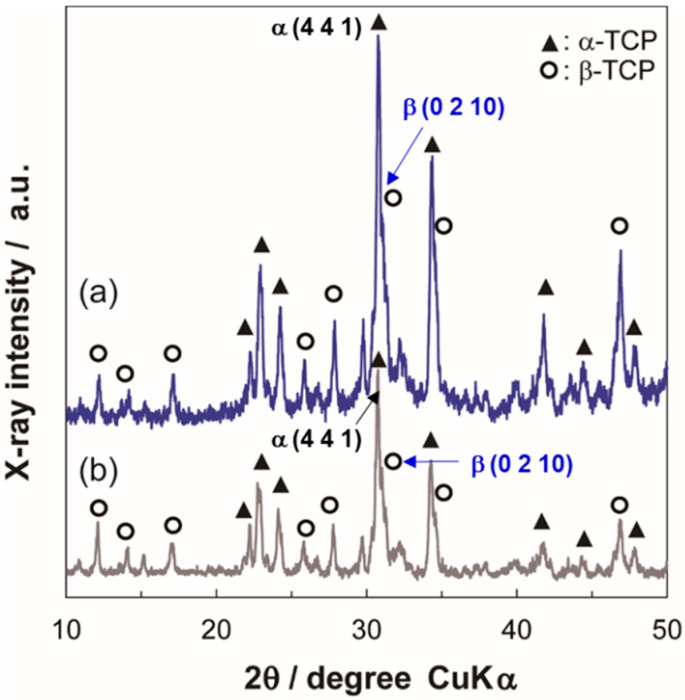
Characterization of as-prepared powders synthesized by ultrasonic spray-pyrolysis (USSP). X-ray diffractometry (XRD) patterns of washed sample powders of (**a**) Silver-containing tricalcium phosphate (Ag-TCP(0)) and (**b**) Ag-TCP(5).

**Figure 2 ijms-21-03745-f002:**
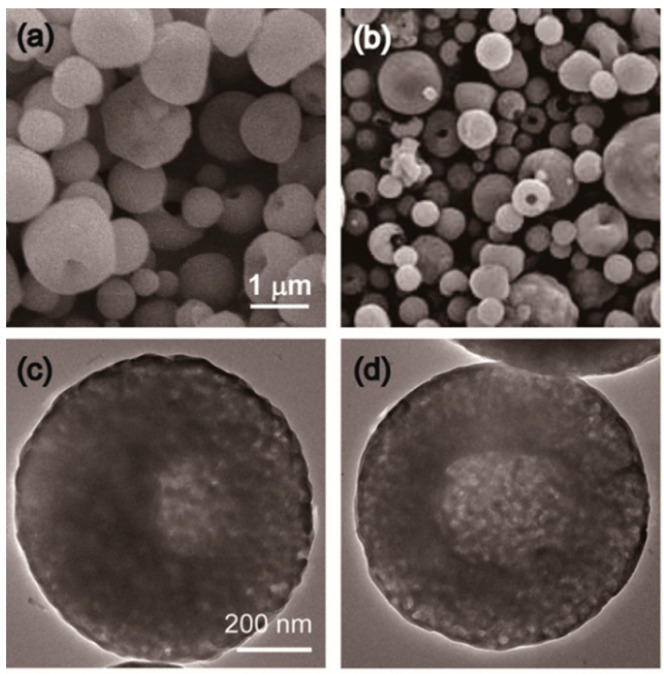
SEM and TEM micrographs of Ag-TCP powders. SEM images of washed sample powders of (**a**) Ag-TCP(0) and (**b**) Ag-TCP(5). Bar indicates 1 μm. High-resolution of TEM images of washed sample powders of (**c**) Ag-TCP(0) and (**d**) Ag-TCP(5). Bar indicates 200 nm.

**Figure 3 ijms-21-03745-f003:**
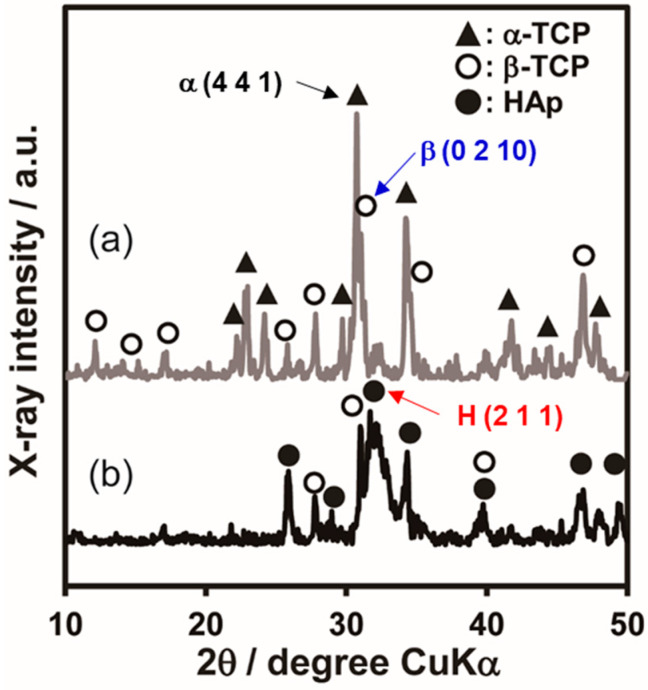
Phase analysis of Ag-TCP(5) powder and cement. XRD patterns of (**a**) Ag-TCP powder and (**b**) Ag-TCP cement.

**Figure 4 ijms-21-03745-f004:**
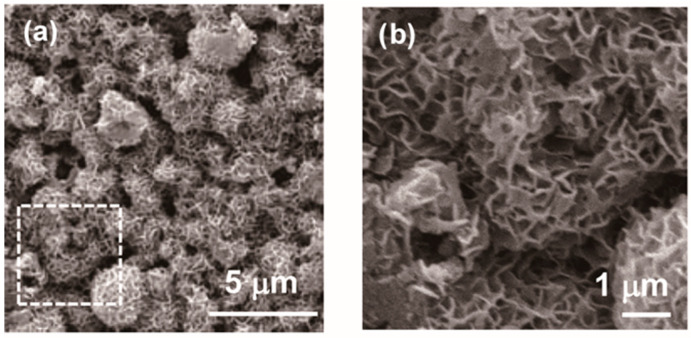
SEM micrographs of Ag-TCP(5) cement. SEM images of the surface of Ag-TCP(5) cement. (**b**) represents high magnification of indicated area. Bars indicate 5 μm (**a**) and 1 μm (**b**), respectively.

**Figure 5 ijms-21-03745-f005:**
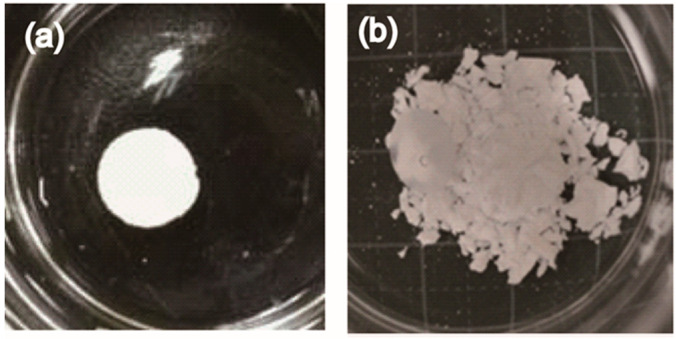
Evaluation of anti-washout property of Ag-TCP cement. Ag-TCP(5) and Ag-HAp(5) cements were immersed into water for 24 h after setting. Ag-TCP(5) cement had anti-washout property (**a**), though Ag-HAp cement collapsed (**b**).

**Figure 6 ijms-21-03745-f006:**
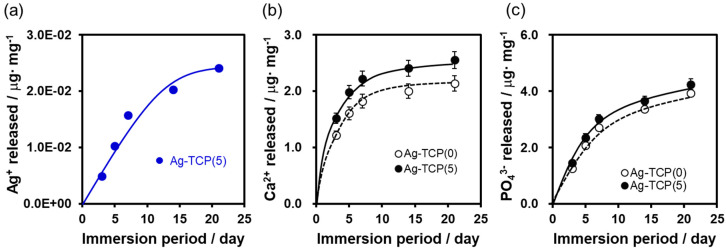
Release kinetics of Ag^+^, Ca^2+^, and PO_4_^3−^ ions from Ag-TCP cements. Release of Ag^+^ ion (**a**), Ca^2+^ ion (**b**), and PO_4_^3−^ ion (**c**) from Ag-TCP cements immersed into 4-(2-hydroxyethyl)-1-piperazineethanesulfonic acid (HEPES) buffer over time.

**Figure 7 ijms-21-03745-f007:**
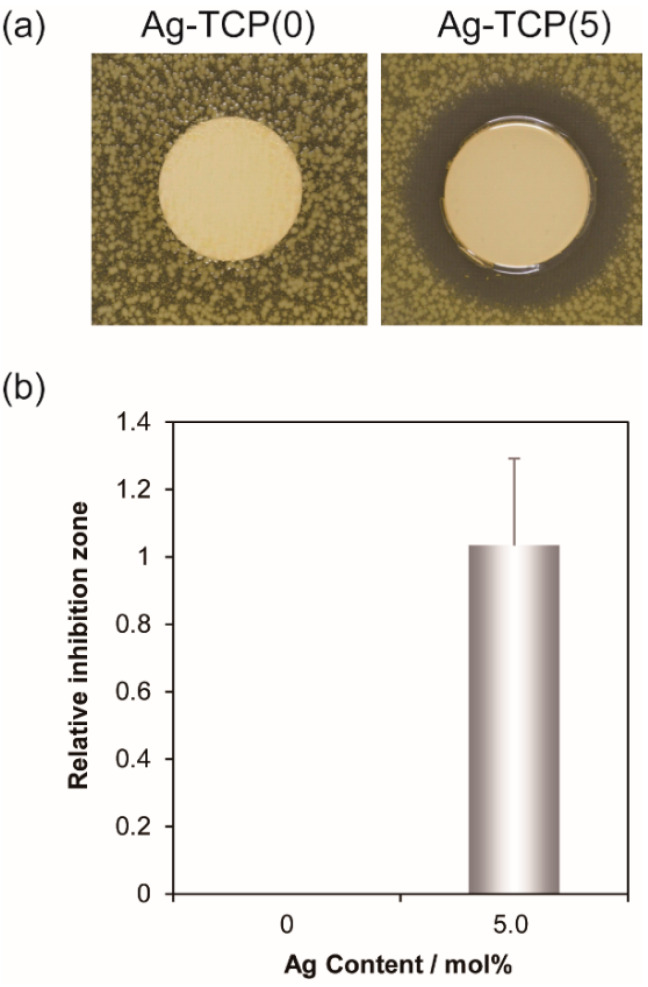
In vitro evaluation of antimicrobial activity by inhibition zone method. Inhibition zone test of Ag-TCP cements: Ag(0) and Ag(5) in *S. aureus* (**a**). Inhibition zone was calculated by Equation (1) (**b**). Error bars indicate standard error of the mean (n = 3).

**Figure 8 ijms-21-03745-f008:**
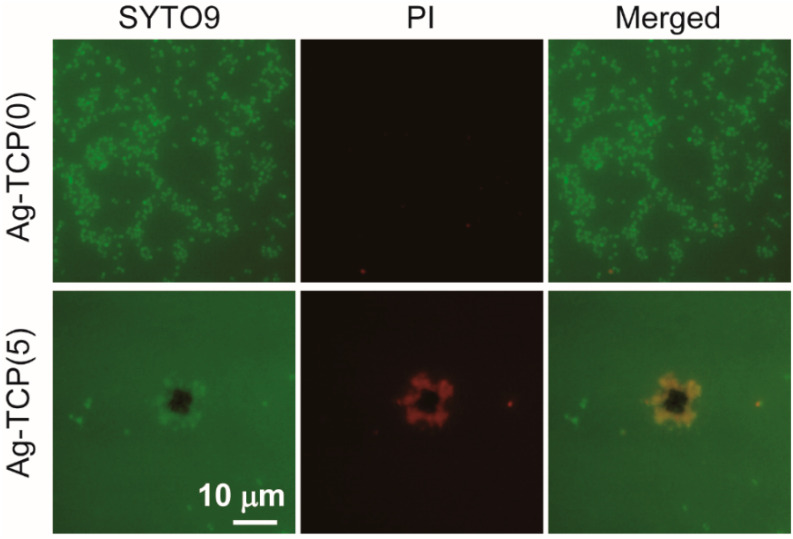
In vitro evaluation of antimicrobial activity by biofilm formation assay. Fluorescence in micrograph (LIVE/DEAD staining) of Ag-TCP(0) and Ag-TCP(5) cements surface after in vitro biofilm formation. Vital bacteria appear green (SYTO9) and dead bacteria appear red (PI). Right panels represent merged image of SYTO9 and PI. Bar indicates 10 μm.

**Figure 9 ijms-21-03745-f009:**
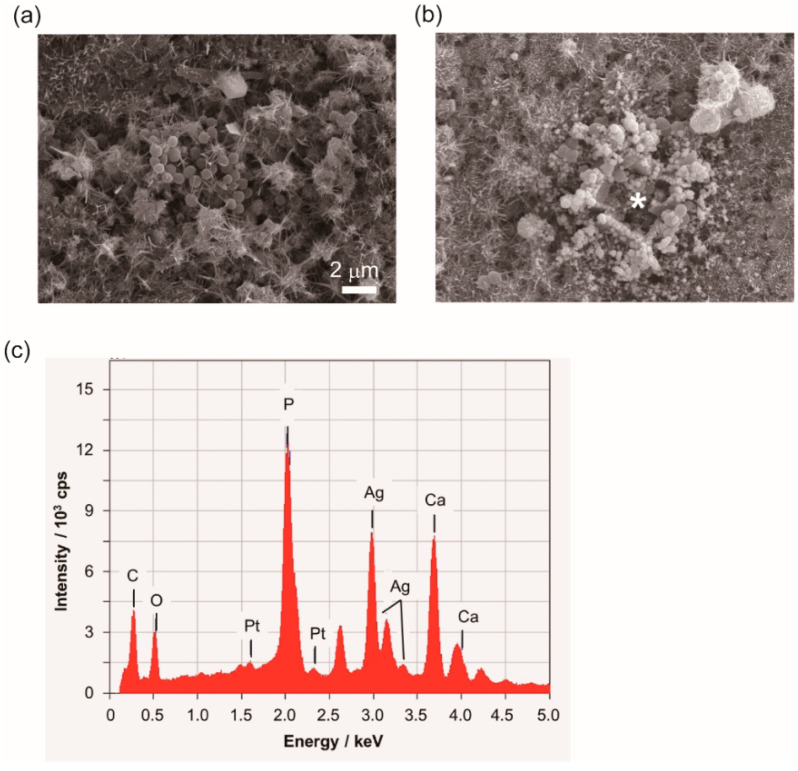
SEM observation of microstructure of cement and morphology of bacteria. Bacterial adherence to the surface of Ag-TCP(0) cement (**a**) and Ag-TCP(5) cement (**b**) at 24 h after culture. Energy dispersive X-ray spectrometry (EDX) spectrum of Ag-TCP(5) cement shows the asterisk in Figure 9b (**c**). Scale bar indicates 2 μm.

**Figure 10 ijms-21-03745-f010:**
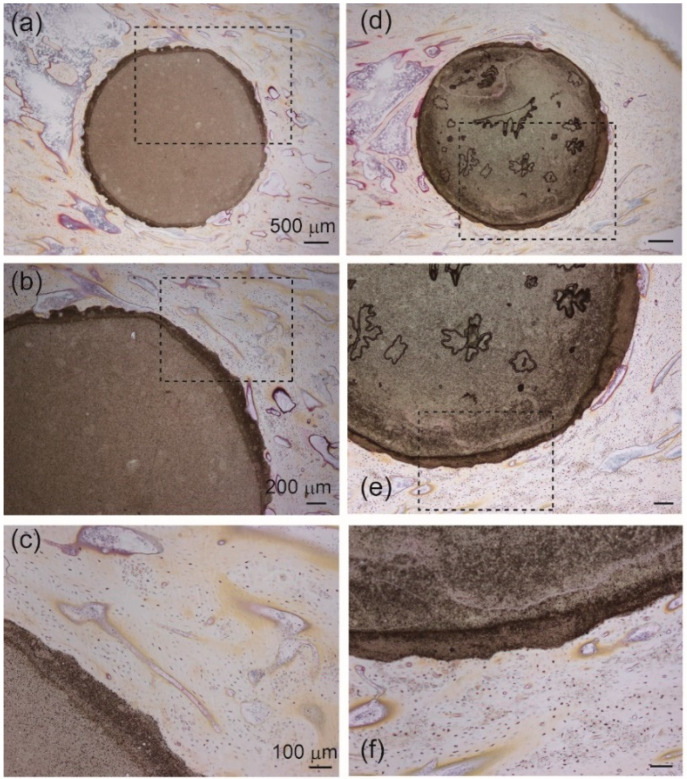
Histological observations (Villanueva bone staining) in Ag-TCP(0) and Ag-TCP(5) cements at 8 weeks after the operation. Left panels (**a**–**c**) showed Ag-TCP(0) cements and right panels (**d**–**f**) represent Ag-TCP(5) cements. (**b**,**e**) show higher magnification of the rectangular area in (**a**,**d**). (**c**,**f**) represent higher magnification of the rectangular area in (**b**,**e**). Scale bars indicate 500 μm (**a**,**d**), 200 μm (**b**,**e**), and 100 μm (**c**,**f**), respectively.

**Table 1 ijms-21-03745-t001:** Quantities of reactants used in the samples.

Sample	Ca(NO_3_)_2_	(NH_4_)_2_HPO_4_	HNO_3_	AgNO_3_	Ag	Ca/P
	mol·dm^−3^	mol·dm^−3^	mol·dm^−3^	mol·dm^−3^	mol%	
Ag-TCP(0)	0.6	0.4	0.4	–	–	1.50
Ag-TCP(5)				3.0×10^−3^	5	

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
