# Peer review of "Bactericidal and Bioresorbable Calcium Phosphate Cements Fabricated by Silver-Containing Tricalcium Phosphate Microspheres"

_ijms, 2020, doi:10.3390/ijms21113745_

Round 1

Reviewer 1 Report

This research focuses on fabricating the anti-bacterial CPC and investigate their biological properties.

The manuscript is well written and it is clear. However, there are some parts that need to be revised.

Comments:

  1. The section “Discussion” of this manuscript should be revised. There are some parts that are irrelevant to this section (e.g. lines 191-205). These information should be moved up to the section “Introduction”. In the section “Discussion”, the authors should only talk about their obtained results and the interpretation of results. Same comment for lines 231-240.
  2. As a suggestion the two sections “Results” and “Discussion” can be merged to one section e.g. “Results and Discussion”.
  3. Authors performed the compressive tests to investigate the mechanical properties (mechanical strength) of their fabricated CPCs. First of all, compressive tests is not clinically relevant. Researchers usually perform this test because it is easier (experimental challenges) to perform compared to the more relevant clinically tests like bending and tensile tests.
  • Why authors did not decide to go for bending and tensile tests? In any case, in some part of the paper it should be mentioned that for more accurate analysis of mechanical properties the bending and tensile properties of CPCs should be investigated. A reference should be added here :
  1. Paknahad, N. W. Kucko, S. C. Leeuwenburgh, L. J. Sluys, Experimental and numerical analysis on bending and tensile failure behavior of calcium phosphate cements, Journal of the Mechanical Behavior of Biomedical Materials 103 (2020) 103565.

  • As it is presented in this paper, the compressive strength of fabricated CPC is around 18 MPa. This is clearly lower than the compressive strength of cortical bone (95-230 MPa). The authors are expected to compare their obtained compressive strength value with previously published values for different type of CPCs and specially with the compressive strength values of cortical bone and native cancellous bone.
  • If the specimens after executing the tests are yet available, it is good to add an image of the fractured specimen after performing the compressive strength and also one average force-displacement curve to show the crack propagation pattern and also the brittleness behavior of the specimens in their force-displacement curve.
  • Did authors check the Weibull modulus for their mechanical test results? How scatter are the compressive test results are? What is the standard deviation there ? How many specimen are tested?! A table containing this information (number of specimen, maximum load for each sample, maximum extension for each sample, compressive modulus) should be added after line 300.
  • At the end of lines 116, 117, 253 and 255 where authors talk about the poor mechanical properties of CPCs, a reference should be added:
  1. Paknahad, N. W. Kucko, S. C. Leeuwenburgh, L. J. Sluys, Experimental and numerical analysis on bending and tensile failure behavior of calcium phosphate cements, Journal of the Mechanical Behavior of Biomedical Materials 103 (2020) 103565.

Author Response

Responses to Reviewer;

To Reviewer #1: Thank you for careful review and valuable comments on our manuscript. Let me respond to several specific comments by point-to-point manner, as follows.

  1. The section “Discussion” of this manuscript should be revised. There are some parts that are irrelevant to this section (e.g. lines 191-205). These information should be moved up to the section “Introduction”. In the section “Discussion”, the authors should only talk about their obtained results and the interpretation of results. Same comment for lines 231-240.

Thank you for appropriate comments. According to reviewer’s comments, some sentences were moved up to the section “Introduction”.

  1. As a suggestion the two sections “Results” and “Discussion” can be merged to one section e.g. “Results and Discussion”.

Responding to reviewer’s suggestion, “Results” and “Discussion” were merged in revised manuscript.

  1. Authors performed the compressive tests to investigate the mechanical properties (mechanical strength) of their fabricated CPCs. First of all, compressive tests is not clinically relevant. Researchers usually perform this test because it is easier (experimental challenges) to perform compared to the more relevant clinically tests like bending and tensile tests.

Why authors did not decide to go for bending and tensile tests? In any case, in some part of the paper it should be mentioned that for more accurate analysis of mechanical properties the bending and tensile properties of CPCs should be investigated. A reference should be added here :

  1. Paknahad, N. W. Kucko, S. C. Leeuwenburgh, L. J. Sluys, Experimental and numerical analysis on bending and tensile failure behavior of calcium phosphate cements, Journal of the Mechanical Behavior of Biomedical Materials 103 (2020) 103565.

Thank you for specific comments for the investigation of the mechanical properties.

The reason we chose compressive strength test is that this cement would be mainly applied to vertebral body compression fractures. It is desirable that CPCs can match the host bone (cancellous bone) in such mechanical properties as strength. The compressive strength of CPC in this study was as same level as cancellous bone.

In terms of materials, CPC is a porous material even after hardening, so compressive strength test is believed to be more suitable than bending and tensile tests. In fact, compressive strength tests have been employed in numerous studies [1]. However, as you point out, we should perform the more relevant clinically tests like bending and tensile tests in future work. Thank you for valuable comment.

In addition, according to reviewer’s comment, we added a reference.

  1. As it is presented in this paper, the compressive strength of fabricated CPC is around 18 MPa. This is clearly lower than the compressive strength of cortical bone (95-230 MPa). The authors are expected to compare their obtained compressive strength value with previously published values for different type of CPCs and specially with the compressive strength values of cortical bone and native cancellous bone.

CPCs has been used in vertebroplasty and kyphoplasty treatment. The aim of this study is to fabricate Ag-TCP cement and apply it to vertebral body compression fractures. Although it is not practically as strong as cortical bone (80-200MPa), it is strong enough to fill the cancellous bone (several MPa).

Previously, Martin and Brown reported CPC with high compressive strength (129-174 MPa). However, the CPC consisted of HAp phase resulting slow resorption and inhibition of vascularization [2]. To improve the mechanical characteristics of CPCs by incorporating specific additives, JH Jang and his colleagues fabricated CPC with genetically engineered elastin-like polypeptides [3]. The CPC showed increased compressive strength by addition of elastin-like polypeptides (about 16 MPa which was >10-fold higher than DW). CPC has a wide variety of types, and strength differs depending on the raw materials and production conditions, so it is difficult to simply compare them. We believe that CPCs are biologically interesting due to their potential to be replaced by new bone. Therefore, we will have to improve their material properties, such as mechanical property and biological property.

  1. If the specimens after executing the tests are yet available, it is good to add an image of the fractured specimen after performing the compressive strength and also one average force-displacement curve to show the crack propagation pattern and also the brittleness behavior of the specimens in their force-displacement curve.

Thank you for appropriate comments about crack propagation pattern and force-displacement curve. It is difficult to do this time, so we would like to try to test them next time.

  1. Did authors check the Weibull modulus for their mechanical test results? How scatter are the compressive test results are? What is the standard deviation there ? How many specimen are tested?! A table containing this information (number of specimen, maximum load for each sample, maximum extension for each sample, compressive modulus) should be added after line 300.

Thank you for appropriate comments about mechanical test. To respond the reviewer’s comment, we added the information as text in the manuscript. In our work, we cannot discuss mechanical properties sufficiently. Therefore, we have to perform more detailed analyses including bending and tensile tests are needed.

  1. At the end of lines 116, 117, 253 and 255 where authors talk about the poor mechanical properties of CPCs, a reference should be added:
  2. Paknahad, N. W. Kucko, S. C. Leeuwenburgh, L. J. Sluys, Experimental and numerical analysis on bending and tensile failure behavior of calcium phosphate cements, Journal of the Mechanical Behavior of Biomedical Materials 103 (2020) 103565.

According to reviewer’s comment, we added the reference in the text.

References

  1. Liu, C.; He, H., Developments and Applications of Calcium Phosphate Bone Cements. Springer Singapore: 2017.
  2. Martin, R. I.; Brown, P. W., Mechanical properties of hydroxyapatite formed at physiological temperature. Journal of Materials Science: Materials in Medicine 1995, 6, (3), 138-143.
  3. Ambard, A. J.; Mueninghoff, L., Calcium Phosphate Cement: Review of Mechanical and Biological Properties. 2006, 15, (5), 321-328.

Reviewer 2 Report

In my humble opinion, this manuscript may be accepted for publication as is.

Author Response

I wish sincerely to express my gratitude for your having kindly proofread our manuscript.

Reviewer 3 Report

The concept of this work is the same with the previous article published in Ref 21. However there is a little novelty since this work is on the cement particles. My suggestion is acceptance after major revision

Comments:

  1. The Materials and Methods part is very similar with the counterpart published in Ref 21.
  2. Lines 117-120: Show the values of compressive strength mean +/- SD for the pure and Ag loaded samples
  3. Table 1: How composition % was calculated using XRD ?
  4. Add Miller indices on some characteristic peaks of a-TCP. b-TCP and HAp
  5. Show TEM images with Ag loaded nanoparticles
  6. Explain the conditions under which a-TCP was formed. It is well known that a-TCP is formed at temperatures higher than 1125 oC.

Author Response

Responses to Reviewer;

To Reviewer #3: Thank you for careful review and valuable comments on our manuscript. Let me respond to several specific comments by point-to-point manner, as follows.

  1. The Materials and Methods part is very similar with the counterpart published in Ref 21.

In our previous report (Ref#26), we prepared Ag-HAp powders and demonstrated their antimicrobial properties. However, Ag-HAp disc did not have the ability of self-hardening and washout resistence. Therefore, we focused on the self-hardening, washout resistance, and bactericidal properties in this study. As you point out, this paper is similar to the previous one, however; to overcome these limitations was the novelty in this paper.

  1. Lines 117-120: Show the values of compressive strength mean +/- SD for the pure and Ag loaded samples

Thank you for appropriate comment. Responding to reviewer’s comment, we have added data.

  1. Table 1: How composition % was calculated using XRD ?

As you stated, composition % in Table 1 was calculated using XRD.

The crystal phase was identified with respect to the JCPDS reference patterns for a-, b-TCP and HAp (#09-0348, #09-0169, and #9-0432). Each content in the powders or cements was calculated using the typical peaks of a-TCP (2q=30.74°), b-TCP (2q=31.03°) and HAp (2q=31.78°).

  1. Add Miller indices on some characteristic peaks of a-TCP. b-TCP and HAp

In accordance with reviewer’s comment, we have added Miller indices in Figures.

  1. Show TEM images with Ag loaded nanoparticles

TEM image of Ag-TCP microsphere was shown in Fig. 2d. As for Ag nanoparticles, we could not observe nanoparticle in these powders. In our previous study, numerous Ag nanoparticles are attached to the surface of only Ag(20)-HAp powders. These data showed that Ag nanoparticles were distributed in samples with Ag contents of more than 20 mol%.

  1. Explain the conditions under which a-TCP was formed. It is well known that a-TCP is formed at temperatures higher than 1125 oC.

Thank you for appropriate comments. As you stated, a-TCP is usually prepared at temperatures above 1125 °C. In this study, we used ultrasonic spray-pyrolysis technique. Using this technique, metastable a-TCP was formed when temperature of powders being produced in the reaction zone was lower than the b to a phase transformation temperature [1]. Additionally, the stable a-TCP could be obtained at spray-pyrolysis temperatures above 1125 °C.

  1. Inoue, S.; Kobayashi, M.; Ono, A., Characterization of Tricalcium Phosphate Powders Prepared by Spray-Pyrolysis Technique. 1988, 96, (1110), 182-185.

Round 2

Reviewer 3 Report

Comment 1

If the Materials and Methods remain very similar with the previous article the authors probably will have plagiarism problems

Comment 3

The authors did not answer my comment. Which formulation used to do quantitative analysis of the three-component mixture. How many times the measurement repeated? Did they prepared standard mixtures with known amounts? What is the SD of their measurements ?

 If the authors just measured the relative intensity of diffraction peaks in one diffractogram to calculate the % amount the calculation is wrong

Another comment is also that the b-TCP does not precipitate in the reactions and the amount is expected to be the same in the powder and in the cement. However the amount is 34 and 47 % respectively. Is this amount expected in the cement?

Comment 5

I understand that there are no images showing Ag particles attached on TCP.

However, the answer that  Ag nanoparticles attached on the surface of Hap (which is a different work) were observed in a previous study is unrealistic.

Round 3

Reviewer 3 Report

The answers are not thorough however they are adequate